

# Enhancing physical activity in older type 2 diabetic adults through remote patient monitoring: a pre-post study in Taiwan

Cheng-Fu Lin[1,2,3,4], Hui-Min Chang[1], Chiann-Yi Hsu[5], Chao-Tung Yang[6,7] and Shih-Yi Lin[1,3,4,8,9]

[1] Center for Geriatrics & Gerontology, Taichung Veterans General Hospital, Taichung, Taiwan
[2] Health Management Center, Taichung Veterans General Hospital, Taichung, Taiwan
[3] Geriatrics and Gerontology Research Center, College of Medicine, National Chung Hsing University, Taichung, Taichung, Taiwan
[4] Department of Post-Baccalaureate Medicine, College of Medicine, National Chung Hsing University, Taichung, Taichung, Taiwan
[5] Biostatistics Task Force, Taichung Veterans General Hospital, Taichung, Taiwan
[6] Department of Computer Science, Tunghai University, Taichung, Taiwan
[7] Research Center for Smart Sustainable Circular Economy, Tunghai University, Taichung, Taiwan
[8] Institute of Clinical Medicine, School of Medicine, National Yang Ming Chiao Tung University, Taipei, Taiwan
[9] Division of Endocrinology and Metabolism, Department of Internal Medicine, Taichung Veterans General Hospital, Taichung, Taiwan

Corresponding authors
Chao-Tung Yang, ctyang@thu.edu.tw
Shih-Yi Lin, sylin@vghtc.gov.tw

## ABSTRACT

**Background.** Type 2 diabetes (T2D) is a growing concern among older adults, increasing the risk of frailty, and functional decline. In Taiwan, the convergence of population aging and high diabetes prevalence calls for innovative care strategies. This study evaluated the effectiveness of incorporating wearable step-count devices into the diabetic pay-for-performance (P4P) program to enhance physical activity and explore associations with related health outcomes.

**Methods.** This prospective, single-arm interventional study was conducted from February to September 2023 at a medical center in central Taiwan. T2D participants in P4P who were able to use smart phone were enrolled. At baseline, comprehensive geriatric assessment was performed to measure participants' physical, mental functions and nutritional status. Daily step data were collected *via* Garmin trackers and synced automatically. Participants received weekly remote feedback from diabetes educators to encourage adherence in 2 months. The Wilcoxon signed-rank test assessed changes in step counts over time, and Spearman's rank correlation examined associations with baseline health indicators. An association of daily step counts with metabolic controls factors, biochemical data, disease severity, functional performance, frailty, nutritional and mood were analyzed.

**Results.** The study involved 66 participants, median age 72 years, with 24 males (36.4%) and 42 females (63.6%). Metabolic indicators showed fasting plasma glucose at 110.0 mg/dL (interquartile range, IQR: 97.0–137.5) and hemoglobin A1c at 6.1 (IQR: 5.7–7.2). Additionally, low-density lipoprotein was 86.5 mg/dL (IQR: 67.3–104.5), and triglycerides were 98.5 mg/dL (IQR: 76.8–139.8). Urine albumin-creatinine ratio was 15.3 (IQR: 7.6–84.9), and estimated glomerular filtration rate (eGFR) was 70.7 mL/min/1.73 m$^2$ (IQR: 48.9–78.1). Functional capacity varied, with 47.0% having
low muscle strength and 92.0% showing low physical performance. 15.2% showed symptoms of depression. Malnutrition and frailty were observed in 6.1% and 13.6%, respectively. Median daily steps significantly increased from 1,560.8 (IQR: 955.9–3,301.5) in week 1 to 2,652.9 (IQR: 1,271.8–4,139.3) in the final week ($p < 0.001$). Higher daily step counts were positively correlated with physical and nutritional status and negatively correlated with age, depressive symptoms, and frailty. Remote monitoring led to a significant and consistent increase in daily step counts across all tracking periods ($p < 0.001$).

**Conclusions**. The study found that digital mobile health monitoring improved daily step counts over time in older diabetic patients, and baseline physical functions, and nutritional status were related to the changes. Whether incorporating this wearable technology into diabetes education program improves long metabolic controls needs further researches.

# INTRODUCTION

Diabetes is a growing global health concern, affecting an estimated 537 million adults worldwide in 2021, with projections indicating a rise to 783 million by 2045 (*Sun et al., 2022*). Taiwan faces similar challenges, as its aging population and lifestyle-related risk factors contribute to an increasing prevalence of diabetes (*Tsai et al., 2021*). Diabetes is a metabolic disorder characterized by chronic hyperglycemia and impaired insulin function, posing significant global health challenges, particularly for older adults, by increasing the risk of premature death, functional disability, muscle loss, and coexisting conditions such as hypertension, coronary heart disease, and stroke (*ElSayed et al., 2023*). Beyond traditional vascular and neuropathic issues, diabetes in older adults is associated with physical and mental disabilities, frailty, which reduces physiological reserve and increases the risk of hospitalization and mortality, and cognitive dysfunction, which complicates disease management (*Sinclair & Abdelhafiz, 2020*; *Subramanian, Vasudevan & Rajagopal, 2021*; *Weng et al., 2023*). Additionally, the disease significantly elevates the risk of advanced dependence and disability, with affected individuals facing a two- to threefold higher risk of impaired daily activities and mobility (*Lin, Liu & Lin, 2022*).

Regular physical activity improves glycemic control in type 2 diabetes (T2D) and may also reduce the risks of physical frailty, Alzheimer's disease, and other dementias (*Colberg et al., 2016*; *ElSayed et al., 2023*; *Nickerson & Shade, 2021*; *Tsujishita, Nagamatsu & Sanada, 2023*). Furthermore, exercise has the potential to boost functional capacity, fitness and the overall health-related quality of life in people with T2D (*Fritschi et al., 2017*). Daily step counts, including the widely known 10,000 steps target, are often employed for the promotion of physical activity (*Del Pozo-Cruz et al., 2022a*; *Del Pozo Cruz et al., 2022b*). In diabetes care, remote patient monitoring (RPM) systems can apply the nudge theory by setting default activity reminders or step count targets that patients must actively opt out of, thereby gently encouraging healthier behavior (*Kwan et al., 2020*; *Ernsting et al., 2017*).

Additionally, wearable trackers, smartphone applications, and remote monitoring devices provide valuable support for older adults seeking to adopt healthier lifestyles. However, most studies have focused on their acceptance rather than their long-term impact on lifestyle changes (*Zheng et al., 2020*).

The pay-for-performance (P4P) program, an initiative widely adopted in Taiwan, incentivizes healthcare providers to enhance diabetes care quality and patient outcomes (*Lee et al., 2019*). While physical activity benefits type 2 diabetes management, the effects of short-term step count interventions on glycemic outcomes remain unclear. This study aims to examine the impact of the RPM program integrating wearable step-count devices within Taiwan's diabetic P4P framework. The first objective of the study was to assess older adults with T2D in a community setting, focusing on the relationship between their physical and cognitive functions, nutritional status, and daily step counts. While the secondary objectives were to continuously monitor step counts using wearable trackers and explore their associations with baseline factors. Though focused on correlations rather than causality, the study offers insights into how increased physical activity may influence health outcomes.

## MATERIALS & METHODS

### Study design
This prospective study was conducted at a medical center in central Taiwan from February 1, 2023, to September 30, 2023. All participants were fully informed about the study and provided written informed consent. The study was approved by the Institutional Review Board (IRB) of Taichung Veterans General Hospital (IRB number: CE21533B) and was carried out in strict adherence to the study protocol and the principles outlined in the Declaration of Helsinki. The study flowchart is presented in Fig. 1.

### Participants
Participants having been given a diagnosis of T2D and participating in P4P program were eligible for enrollment. Participants were recruited through physician referrals and self-selection using a convenience sampling approach. All subjects demonstrated familiarity with using the internet and smart devices, owned a smartphone, and had both the ability and willingness to complete the study procedures. After enrollment and training, our study was designed to be completed within a two-month intervention at our diabetes outpatient clinic. The exclusion criteria included participants with cognitive impairment, those who refused evaluation, individuals unable to complete the two-month study period for any reason, those participating in another clinical trial, and individuals unwilling to discontinue other interventions for the study duration.

### Interventions
In our study, the participants wore wrist Garmin activity trackers, which through Bluetooth automatically transmitted data periodically whenever the device was in the proximity of a smartphone having the configured tracking mobile application. This application typically relied on the built-in sensors of wearable devices to accurately track movement
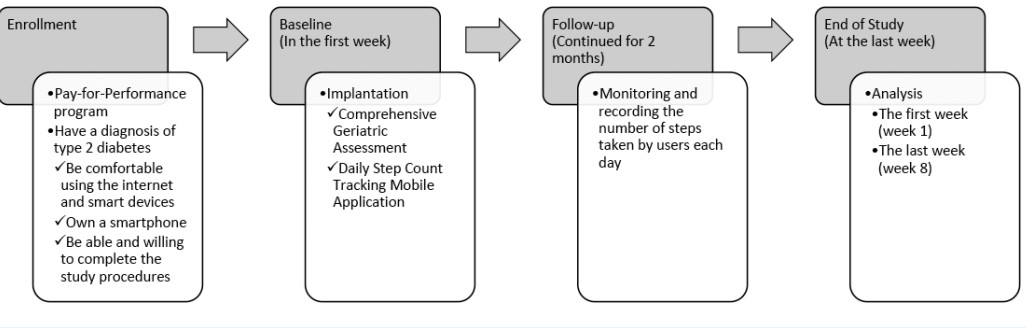

**Figure 1 Flowchart of the study.**

and count steps. Prior studies validated the reliability of step tracking in older adults, demonstrating strong accuracy (*Evenson & Spade, 2020*; *Tedesco et al., 2019*). Simplified device interfaces and technical support improved usability and data validity within this population (*Evenson & Spade, 2020*; *Farivar, Abouzahra & Ghasemaghaei, 2020*; *Tedesco et al., 2019*). The application seamlessly interacted with a centralized platform to synchronize user data, including basic information, vital signs and step counts. Additionally, the platform offered supplementary features, such as collective step count monitoring for the educators. This feature empowered the educators to oversee group performance, gain insights into collective step counts, and provide timely support to participants. In situations where participants failed to input data automatically, the platform took proactive measures by sending notifications to both participants and managers *via* the application, ensuring all involved parties remained informed. Alternatively, the educators had the option to reach out to participants *via* their phone to check on their status if necessary. Each participant received a personal step-count device, with the intervention delivered at an individual level. The research team, consisting of physicians and clinical staff, provided instructions and technical support. The intervention was conducted in participants' homes to facilitate continuous remote monitoring. The two-month study required participants to wear the device daily, chosen for feasibility and evidence that behavioral changes, like increased physical activity, can develop within this period (*Lally et al., 2010*). The two-month intervention began with a one-week baseline period, during which participants carried their mobile phones during waking hours and maintained their usual activity levels. Throughout the study, educators encouraged participants to monitor their own step counts while also providing feedback *via* the platform to promote physical activity based on the previous week's data. Step counts were averaged weekly, requiring data from at least three of five working days. Monthly averages were similarly calculated to ensure adherence.

## Outcomes

The primary outcome was the average daily step count recorded by wearable devices over the two-month intervention period. Secondary outcomes included correlations between the baseline diabetic P4P program, comprehensive geriatric assessment (CGA) parameters and changes in step counts over time. Data collection methods included automatic

synchronization of step count data *via* wearable devices and in-person assessments using standardized CGA protocols administered by trained research staff.

## The diabetic pay-for-performance program

The diabetic P4P program was implemented during the study period (*Lee et al., 2019*), which included a series of recommended examinations, including hemoglobin A1c (A1C), fasting plasma glucose (FPG), triglyceride, cholesterol, low-density lipoproteins (LDL), alanine transaminase, creatinine, and urine albumin-creatinine ratio. Estimated glomerular filtration rate (eGFR) was determined for Taiwanese adults using the Modification of Diet in Renal Disease equation (*Chen et al., 2014*). Furthermore, diabetic education sessions were provided to enhance care and management (*Lee et al., 2019*).

## Comprehensive geriatric assessment

The CGA encompasses a range of variables, including age, gender, body mass index and educational level (*Chen et al., 2010*). Functional capacity of our participants was assessed using the Barthel index (BI) for evaluating activities of daily living (ADL), as well as the Lawton Instrumental ADL (IADL) scale. The BI assesses a person's ability to perform tasks such as bathing, dressing, feeding, toileting, transferring and continence, with a lower score indicating a higher level of disability and thus a greater need for assistance with these fundamental daily activities (*Chen et al., 2010*). The IADL encompasses more complex tasks crucial for independent living, including meal preparation, managing finances, shopping, housework, and using the telephone and transportation, with a lower score indicating a greater need for assistance or support in these intricate daily living tasks (*Graf, 2008*). Mood was evaluated using the five-item geriatric depression scale (GDS-5), where a score of 2 or greater indicates the presence of depressive symptoms (*Lin, Liu & Lin, 2022*). Nutritional status was determined by the mini-nutritional assessment-short form (MNA-SF), where a score below 12 indicates a risk of malnutrition (*Rubenstein et al., 2001*). Possible sarcopenia was evaluated through the measurement of hand grip strength (HGS). For males, a weight less than 28 kg is considered abnormal, while for females, it is 18 kg (*Chen et al., 2020*). Lower extremity mobility was assessed using the 6-meter walking speed (6MWS) test, where a value below 1 m/s suggests low physical performance (*Chen et al., 2020*). Frailty was assessed using Fried frailty phenotype (FFP), which considers the presence of fatigue, unintentional weight loss, weakness, sluggishness and low physical activity. Individuals meeting any two of these criteria are classified as prefrail, while meeting more than three criteria categorizes them as frail (*Fried et al., 2001*).

## Statistical analyses

Non-parametric tests were used due to data deviation from normal distribution. Continuous variables were shown as median and interquartile range (IQR, 25%–75%), while categorical data were expressed as numbers and percentages. Group comparisons were conducted using the Mann–Whitney $U$ test for continuous variables, and the Fisher's Exact test and Chi-Square test for categorical variables. A Wilcoxon signed-rank test was used to compare the paired data, and a *post hoc* power analysis was performed based on an effect size (dz) of 0.4368, an alpha level of 0.05. It was determined that a total sample

size of 66 could attain a calculated power of 92.7%. The Spearman's rank correlation coefficient was assessed for the relationship between the tracking of daily step count as the independent variables, and metabolic control factors, biochemical data, disease severity, functional performance, frailty, nutritional and mood as the dependent variables. To analyze relevant factors of daily step counts, participants were divided into two groups based on the comparison of their step count data from the first and last weeks. The baseline step count was established using the data from the first week. If a participant's step count showed a decline in the last week compared to the first week, they were classified into the declined group. Conversely, if their step count improved or remained the same, they were classified into the improved group. Multiple comparisons were adjusted using the Benjamini–Hochberg procedure to control the false discovery rate. Statistical analyses were performed using SPSS version 22.0 (SPSS Inc., Chicago), with a two-tailed $p$-value of $<0.05$ deemed statistically significant.

## RESULTS

A total of 66 participants enrolled in the study and successfully completed the follow-up. Table 1 outlines the baseline characteristics of the participants, which included a median age of 72.0 years and a gender distribution of 24 (36.4%) males and 42 (63.6%) females. A primary school educational level was predominant, constituting 21 (31.8%) of the participants. The data illustrated efficient diabetes management, as evidenced by FPG at 110.0 mg/dL (IQR: 97.0–137.5) and A1C at 6.1% (IQR: 5.7−7.2). Assessments indicated well-controlled liver and kidney functions. The lipid profile portrayed effective management, with LDL at 86.5 mg/dL (IQR: 67.3–104.5). The functional capacity of the participants displayed diverse levels of independence. Specifically, 31 (47.0%) participants exhibited low muscle strength, and a significant majority, 46 (92.0%), showed signs of low physical performance. Additionally, depression was noted in 10 (15.2%) subjects. Malnutrition was observed in four (6.1%), with nine (13.6%) identified as frail based on the FFP. Monitoring daily step counts during designated periods revealed the following medians (IQR): the first week −1,560.8 (955.9–3,301.5), and the last week −2,652.9 (1,271.8–4,139.3). The average daily step count increased from 2,535.8 ± 2,350.3 steps during the first week to 2,950.1 ± 2,388.2 steps during the last week.

Table 2 illustrates how specific factors connected with daily step counts. Age demonstrated a moderately negative correlation with daily steps, suggesting that as age increased, steps generally decreased ($p < 0.003$). The eGFR is positively correlated with daily step counts, it implies that individuals who have better kidney function tend to engage in more physical activity. Daily activities such as ADL, IADL and HGS showed positive correlations with daily steps, indicating that exhibiting better abilities was linked to more steps (all $p$-values were significant). In contrast, the GDS-5 showed a negative correlation with daily steps, with $p$-values consistently below 0.030, indicating a reduction in depression symptoms. The MNA-SF demonstrated a positive correlation with daily steps, with $p$-values consistently below 0.002, reflecting an enhancement in nutritional status. Notably, the FFP exhibited a robust negative correlation with daily steps, with $p$-values consistently below 0.001, signifying a substantial decrease in frailty.

**Table 1  Baseline characteristics of the participants.**

|  | Median/N | IQR/% |
|---|---|---|
| Age (years) | 72.0 | (68.0–78.0) |
| Gender |  |  |
| Male | 24 | (36.4%) |
| Female | 42 | (63.6%) |
| Body mass index (kg/m$^2$) | 24.9 | (22.5–27.3) |
| Educational level |  |  |
| Illiterate | 10 | (15.2%) |
| Literate | 3 | (4.5%) |
| Primary school | 21 | (31.8%) |
| Junior high school | 13 | (19.7%) |
| Senior high school | 10 | (15.2%) |
| University | 9 | (13.6%) |
| Laboratory data |  |  |
| Fasting plasma glucose (mg/dL) | 110.0 | (97.0–137.5) |
| Hemoglobin A1c (%) | 6.1 | (5.7–7.2) |
| Alanine transaminase (U/L) | 18.5 | (14.0–23.8) |
| Creatinine (mg/dL) | 1.0 | (0.8–1.4) |
| Estimated glomerular filtration rate (mL/min/1.73 m$^2$) | 70.7 | (48.9–78.1) |
| Cholesterol (mg/dL) | 157.0 | (137.0–177.0) |
| Low-density lipoproteins (mg/dL) | 86.5 | (67.3–104.5) |
| Triglyceride (mg/dL) | 98.5 | (76.8–139.8) |
| Urine albumin-creatinine ratio | 15.3 | (7.6–84.9) |
| Hemoglobin (g/dL) | 12.7 | (11.5–13.8) |
| Comprehensive geriatric assessment |  |  |
| Barthel index of activities of daily living | 100.0 | (90.0–100.0) |
| Lawton instrumental activities of daily living | 7.5 | (4.0–8.0) |
| Handgrip strength (Kg) | 19.4 | (15–24.2) |
| 6-meter walking speed (m/s) | 0.7 | (0.5–0.8) |
| Five-item geriatric depression scale | 1.0 | (0.0–1.0) |
| Mini-nutritional assessment-short form | 14.0 | (11.0–14.0) |
| Fried frailty phenotype | 0.0 | (0.0–0.3) |
| Tracking daily step counts |  |  |
| The first week (steps) | 1,560.8 | (955.9–3,301.5) |
| The last week (steps) | 2,652.9 | (1,271.8–4,139.3) |

In Fig. 2, the median daily step count significantly increased from 1,560.8 steps (IQR: 955.9–3,301.5) in the first week to 2,652.9 steps (IQR: 1,271.8–4,139.3) in the last week. This increase was statistically significant ($p < 0.001$), indicating a meaningful improvement in physical activity over time. Based upon the change of daily step counts throughout the entire study (Table 3), the participants were categorized into improved and declined groups, and some significant differences were observed between them. While initial analyses suggested that the declined group had higher IADL scores and took more steps in the first week,

**Table 2  The relationship between baseline parameters and the tracking of daily step counts (week 1 and week 8).**

|  | The first week (week 1) | | The last week (week 8) | |
|---|---|---|---|---|
|  | $r_s$ | p value | $r_s$ | p value |
| Age (years) | −0.42 | 0.001 | −0.38 | 0.002 |
| Body mass index(kg/m$^2$) | 0.13 | 0.309 | 0.13 | 0.308 |
| Fasting plasma glucose (mg/dL) | 0.13 | 0.325 | 0.09 | 0.477 |
| Hemoglobin A1c (%) | 0.09 | 0.498 | 0.09 | 0.495 |
| Alanine transaminase (U/L) | 0.11 | 0.409 | 0.11 | 0.423 |
| Creatinine (mg/dL) | −0.26 | 0.047 | −0.24 | 0.072 |
| Estimated glomerular filtration rate (mL/min/1.73 m$^2$) | 0.33 | 0.010 | 0.35 | 0.006 |
| Cholesterol (mg/dL) | 0.08 | 0.587 | 0.03 | 0.814 |
| Low-density lipoproteins (mg/dL) | 0.08 | 0.560 | 0.06 | 0.685 |
| Triglyceride (mg/dL) | −0.16 | 0.237 | −0.21 | 0.136 |
| Urine albumin-creatinine ratio | −0.06 | 0.741 | −0.20 | 0.225 |
| Hemoglobin (g/dL) | 0.17 | 0.228 | 0.19 | 0.178 |
| Barthel index of activities of daily living | 0.42 | <0.001 | 0.45 | <0.001 |
| Lawton instrumental activities of daily living | 0.54 | <0.001 | 0.51 | <0.001 |
| Handgrip strength (Kg) | 0.26 | 0.035 | 0.32 | 0.010 |
| 6-meter walking speed (m/s) | 0.10 | 0.498 | 0.08 | 0.558 |
| Five-item geriatric depression scale | −0.28 | 0.024 | −0.33 | 0.008 |
| Mini-nutritional assessment-short form | 0.37 | 0.002 | 0.39 | 0.001 |
| Fried frailty phenotype | −0.57 | <0.001 | −0.55 | <0.001 |

these findings were not statistically significant after correcting for multiple comparisons, indicating that the observed differences may have occurred by chance.

## DISCUSSION

This study performed in older adults diagnosed with T2D found a correlation between age, physical function, emotional and nutritional status, and one's daily step count. This suggests that daily activity may impact functional maintenance. Our findings suggest that integrating RPM into diabetes management enhances patient engagement, leading to a significant increase in daily step counts over the two-month tracking period.

A holistic approach that combines medical interventions, remote monitoring, virtual consultations and proactive lifestyle choices can significantly enhance the management of health conditions (*Noah et al., 2018*). While the promising results of interventions based on health behavior models and personalized coaching indicate the potential of RPM, addressing the existing gaps remains crucial for its successful implementation (*Alshagrawi & Abidi, 2023*). The remote monitoring approach promotes patient engagement and empowerment (*Rens et al., 2021*). Patients become active participants in managing their cardiovascular health as they receive real-time feedback on their frailty and functional capacity. This not only fosters a sense of control but also encourages an adherence to lifestyle modifications and treatment plans (*Park et al., 2023*; *Rens et al., 2021*; *Wijsman et*

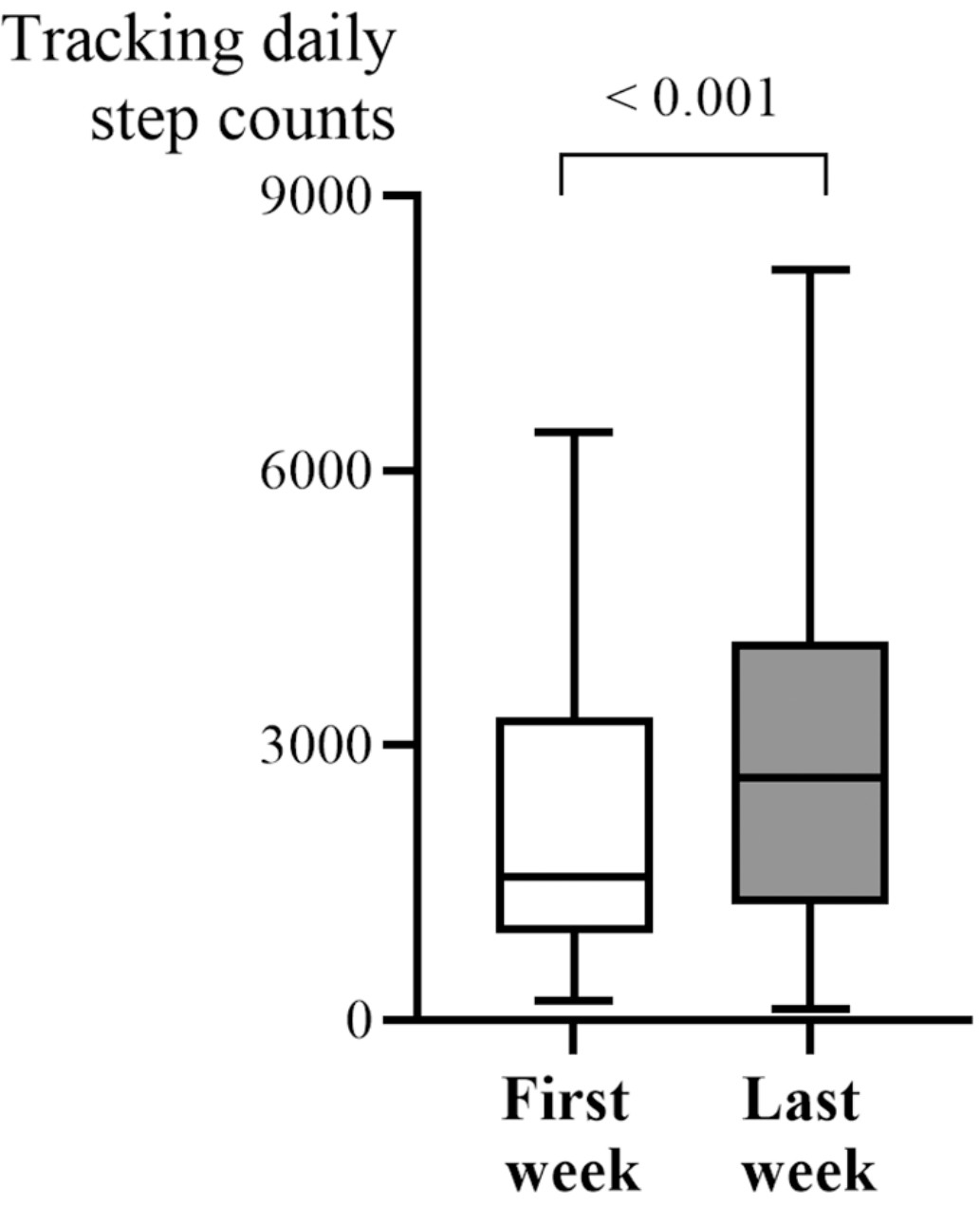

**Figure 2** Significant increase in median daily step count from week 1 to week 8.

*al., 2013*). Our study found evidence that the positive influence of maintaining tracking periods beyond the first week with regards to daily steps suggests that sustained efforts and interventions over time can yield positive outcomes, thus providing a tangible goal for enhancing mobility in the aging population. The integration of lifestyle medicine case manager nurse into healthcare models can significantly improve adherence to lifestyle changes, particularly in older adults (*Cangelosi et al., 2024*; *Chudowolska-Kiełkowska & Małek, 2020*). Our study also included an educator, and the nurse-led intervention aligned

**Table 3  Comparing daily step counts before and after the study.**

| | Improved group (n = 52) | | Declined group (n = 14) | | p value |
|---|---|---|---|---|---|
| | Median/N | IQR/% | Median/N | IQR/% | |
| Age (years) | 73.0 | (68.0–78.0) | 70.5 | (66.8–76.0) | 0.242 |
| Gender | | | | | 0.955 |
| Male | 19 | (36.5%) | 5 | (35.7%) | |
| Female | 33 | (63.5%) | 9 | (64.3%) | |
| Body mass index (kg/m$^2$) | 25.0 | (23.3–27.0) | 23.6 | (21.5–32.2) | 0.667 |
| Educational level | | | | | 0.291 |
| Illiterate | 7 | (13.5%) | 3 | (21.4%) | |
| Literate | 2 | (3.8%) | 1 | (7.1%) | |
| Primary school | 19 | (36.5%) | 2 | (14.3%) | |
| Junior high school | 8 | (15.4%) | 5 | (35.7%) | |
| Senior high school | 9 | (17.3%) | 1 | (7.1%) | |
| University | 7 | (13.5%) | 2 | (14.3%) | |
| Laboratory data | | | | | |
| Fasting plasma glucose (mg/dL) | 109.0 | (96.3–134.5) | 121.0 | (104.0–156.5) | 0.350 |
| Hemoglobin A1c (%) | 6.1 | (5.7–7.1) | 6.3 | (5.7–7.9) | 0.571 |
| Alanine transaminase (U/L) | 18.5 | (14.0–23.8) | 17.5 | (11.8–29.5) | 0.739 |
| Creatinine (mg/dL) | 1.0 | (0.8–1.4) | 0.9 | (0.7–1.2) | 0.235 |
| Estimated glomerular filtration rate (mL/min/1.73 m$^2$) | 68.9 | (48.8–75.4) | 77.9 | (55.7–91.3) | 0.243 |
| Cholesterol (mg/dL) | 158.0 | (139.0–178.0) | 155.0 | (109.5-180) | 0.420 |
| Low-density lipoproteins (mg/dL) | 88.5 | (68.5–109.8) | 85.0 | (57.5–96.8) | 0.345 |
| Triglyceride (mg/dL) | 96.0 | (75.5–142.0) | 102.0 | (80.0–121.0) | 0.871 |
| Urine albumin-creatinine ratio | 13.4 | (7.4–88.5) | 16.0 | (9.1–221.8) | 0.608 |
| Hemoglobin (g/dL) | 12.7 | (11.3–13.6) | 12.8 | (12.4–15.1) | 0.191 |
| Comprehensive geriatric assessment | | | | | |
| Barthel index of activities of daily living | 100.0 | (90.0–100.0) | 100.0 | (87.5–100.0) | 0.750 |
| Lawton instrumental activities of daily living | 7.0 | (4.0–8.0) | 8.0 | (7.5–8.0) | 0.036 |
| Handgrip strength (Kg) | 19.0 | (14.5–24.6) | 21.1 | (15.0–24.5) | 0.730 |
| 6-meter walking speed (m/s) | 0.7 | (0.5–0.8) | 0.7 | (0.5–0.9) | 0.698 |
| Five-item geriatric depression scale | 1.0 | (0.0–1.0) | 1.0 | (0.0–1.0) | 0.952 |
| Mini-nutritional assessment-short form | 14.0 | (11.0–14.0) | 14.0 | (13.3–14.0) | 0.165 |
| Fried frailty phenotype | 0.0 | (0.0–1.0) | 0.0 | (0.0–0.0) | 0.077 |
| Tracking daily step counts | | | | | |
| The first week (steps) | 1,333.3 | (870.5–2,611.3) | 3,285.6 | (2,039.1–5,417.2) | 0.003 |
| The last week (steps) | 2,610.3 | (1,223.6–4,076.1) | 2,716.1 | (1,392.5–4,344.9) | 0.347 |

with existing evidence supporting the critical role of nurses in health promotion and chronic disease management. Participating in physical activity has positive effects on glycemic control, fitness and quality of life in newly diagnosed type 2 diabetic patients (*Nguyen et al., 2023*). In our study, the lack of correlation between baseline HbA1c levels and daily step counts may be attributed to the relatively short intervention period. Additionally, compared to similar studies (*Zheng et al., 2020*), our participants had a lower daily step

count, which was significantly below the commonly held belief that 10,000 steps per day are necessary for maintaining health. However, the integration of the diabetic P4P program may have uniquely motivated participants, as Taiwan's healthcare system incentivizes preventive care more aggressively than fee-for-service models (*Lee et al., 2019*).

Engaging in any level of physical activity offers obvious health benefits when compared to a sedentary lifestyle, as evidenced by numerous studies and meta-analyses showing positive associations, with outcomes such as a reduced risk of diabetes, cardiovascular mortality, and overall mortality being seen (*Hall et al., 2020*; *Smith et al., 2016*). Importantly, a reduced risk of dysglycemia and dementia has been observed even at low levels of daily steps which fall below the commonly ascribed threshold of 10,000 steps per day (*Del Pozo-Cruz et al., 2022a*; *Del Pozo Cruz et al., 2022b*; *Hall et al., 2020*). In our study, despite performing an average of 1,560.8 steps per day, we found there was an association between daily steps and renal, physical functions and emotional status. Several studies have shown that low physical activity levels correlate with chronic kidney disease (CKD) progression, with research indicating that daily step counts decrease as CKD severity worsens (*Zhang et al., 2022*). However, increasing one's physical activity remains challenging, with limited studies showing significant improvements in step counts (*Huang et al., 2024*). In line with previous studies, we found a positive correlation between daily step counts and eGFR in CKD patients. It has been suggested that higher physical activity levels may be linked to better kidney function, although further research is still needed in order to better elucidate this causal relationship.

Previous studies exploring the relationship between objectively measured step counts and depressive symptoms consistently indicate a connection between reduced physical activity and higher levels of depressive symptoms (*Ramsey et al., 2022*). Our study supports the negative correlation with depression symptoms, thus underscoring the importance of addressing mental well-being so as to promote an active lifestyle in older adults.

The association between step counts and frailty underscores the crucial role which regular physical activity plays in maintaining one's health and preventing frailty (*Watanabe et al., 2020*). Numerous studies have demonstrated that an active lifestyle, characterized by higher step counts and overall physical activity, is associated with a lower risk of developing frailty, particularly in aging populations (*Tolley et al., 2021*; *Watanabe et al., 2020*; *Watanabe et al., 2023*). Our study also shows that lower FFP scores, indicative of reduced frailty, were associated with higher daily step counts.

Nutritional status could also be related to physical activities, as adequate nutrition is essential for maintaining physical health (*Papadopoulou et al., 2023*). The relationship between better nutrition and higher levels of physical activity is a well-established and interconnected aspect of overall health and well-being (*Rajabi, Sabouri & Hatami, 2021*). In line with a previous study as well as ours, poor nutritional status was found to be associated with low daily step counts.

The decline in mobility often accompanies the aging process, serving as a natural aspect of growing older (*Rantanen, 2013*). As individuals age, there tends to be a shift in physical abilities, marked by a gradual decrease in mobility (*Rantanen, 2013*). Various age-related factors, such as the loss of muscle mass, joint stiffness, reduced bone density and alterations

in sensory perception, all contribute to this phenomenon of diminishing mobility (*Billot et al., 2020*). Monitoring one's daily step count emerges as a straightforward and direct measure of an individual's physical activity volume (*Amagasa et al., 2021*). Furthermore, in our study, we not only identified a negative correlation between age and step counts among older adults with T2D, but also recognized the importance of considering ADL and IADL. The decline in mobility associated with aging can impact an individual's ability to perform essential daily tasks and activities that require more independence (*Amaral Gomes et al., 2021*).

Our study faces several limitations. Firstly, selection bias is a consideration, as participants were consecutively recruited from a single medical center and were limited to individuals capable of visiting the outpatient department, while those with cognitive impairment were excluded to ensure data reliability. Secondly, the sample size is limited, and the short follow-up period may not be adequate for acquiring a new walking habit and its long-term impact on metabolic markers such as HbA1c. Thirdly, the absence of a control group makes it difficult to determine the extent to which the observed improvement in step counts at the end of the study can be attributed to the intervention. Additionally, residual confounding factors, such as seasonal variations and the Hawthorne effect, may still influence the results. To establish a causal link between daily step counts, cognition, physical functions and nutritional status in older adults with T2D, further longitudinal analyses with a larger and more diverse participant pool, utilizing RPM, are necessary.

## CONCLUSIONS

In summary, this study found that the use of digital mobile health monitoring may help older people with T2D improve their physical activity levels, and factors such as ages, underlying physical, and mental functions, and nutritional status all play crucial roles in influencing step counts over time. Further studies, to elucidate whether by utilizing RPM incorporated into diabetes education program can help promote a more active lifestyle, and thus metabolic control in older individuals with T2D, are warranted.

## ACKNOWLEDGEMENTS

We would like to extend our gratitude to the Biostatistics Task Force of Taichung Veterans General Hospital for their valuable assistance with the statistical analysis performed in this study.

### Funding
The authors received no funding for this work.

### Competing Interests
The authors declare there are no competing interests.

## Author Contributions

- Cheng-Fu Lin performed the experiments, analyzed the data, prepared figures and/or tables, and approved the final draft.
- Hui-Min Chang performed the experiments, prepared figures and/or tables, and approved the final draft.
- Chiann-Yi Hsu analyzed the data, prepared figures and/or tables, and approved the final draft.
- Chao-Tung Yang conceived and designed the experiments, authored or reviewed drafts of the article, and approved the final draft.
- Shih-Yi Lin conceived and designed the experiments, analyzed the data, authored or reviewed drafts of the article, and approved the final draft.

## Human Ethics

The following information was supplied relating to ethical approvals (i.e., approving body and any reference numbers):

The research was approved by The Institutional Review Board of Taichung Veterans General Hospital. (TCVGH-IRB CE21533B).

## Data Availability

The raw measurements are available in the Supplementary File.

## Supplemental Information

Supplemental information for this article can be found online at http://dx.doi.org/10.7717/peerj.19659#supplemental-information.

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
