# Peer review of "Enhancing physical activity in older type 2 diabetic adults through remote patient monitoring: a pre-post study in Taiwan"

_PeerJ, doi:10.7717/peerj.19659_

## Round 0.1 · original submission · Major Revisions

Dear Dr. Lin,

Thank you for your submission to PeerJ.

It is my opinion as the Academic Editor for your article - Enhancing type 2 diabetes management in older adults through remote patient monitoring: The impact of step count data - that it requires some revision.


Warm regards

Reviewer 1 ·

Basic reporting

See the general comments under

Experimental design

See the general comments under

Validity of the findings

See the under general comments

Additional comments

Review: Enhancing Type 2 Diabetes Management in Older Adults through Remote Patient Monitoring: The Impact of Step Count Data
Dear Authors,
First of all, I would like to express my sincere gratitude for allowing me the opportunity to provide my feedback on your manuscript. I found the topic addressed to be extremely interesting and relevant to our field. The research presents several useful and promising insights that could bring significant advancements to our sector. However, after a thorough review, I believe there are some aspects, mainly methodological, that need to be improved and clarified in order to fully appreciate the value of the proposed work. Below, I outline the main areas that could benefit from further development and revision.
Editing: I suggest using the acronym T2D, which has been widely established, as it would make the manuscript easier to read. In line 82, I recommend avoiding the term "mellitus," which is no longer commonly used within the relevant scientific community. As already mentioned in the text (T2D), I suggest the same for the generic term "diabetes."
Title: I recommend specifying the setting (e.g., “Taiwan Population”) and the type of study conducted; this aspect is also lacking throughout the rest of the manuscript.
Abstract: The conclusion section lacks a real interpretation of the clinical implications of the studied phenomenon, along with specific suggestions and proposals for the relevant scientific community.
Keywords: I do not find any keywords listed. If this is an editorial choice, that’s fine, but if it was an oversight, they should be added appropriately.
Introduction: This section could benefit from including an international epidemiological overview and a description of the relevant setting, perhaps reducing the more consolidated clinical content. The objectives are somewhat unclear and expressed in a very general manner. I suggest rewording them as “the main objective of the study was…” and “while the secondary objectives were…”.
Methods: This is the most controversial aspect of the study and certainly deserves more attention. The lack of a structured reporting method, such as the STROBE checklist (doi:10.1016/j.jclinepi.2007.11.008), should be addressed. The inclusion of this checklist as a supplementary file and its citation within the text would be essential for improving the scientific validity and transparency of the study. Additionally, dividing the manuscript into specific sections would aid readers in understanding the study and facilitate the evaluation of the proposed research. Furthermore, the study design is not clearly stated: I believe it to be a prospective cohort study, but this is not immediately apparent to a non-expert reader.
Results: This is certainly the least controversial section of the study. While I would not suggest significant changes as I did with the methods section, I do recommend including the units of measurement for the y-axis in Figure 2.
Discussion: To improve this section, I would suggest expanding the bibliography by comparing other care settings and/or less heterogeneous populations, possibly in the context of Lifestyle Medicine. In this regard, including the input of healthcare professionals who can offer their expertise might prove crucial for the scientific improvement of the discussion. I would like to propose the addition of a figure representing the “Lifestyle Medicine Case Manager Nurses” as a potential component of the proposed model.
Limitations: In my opinion, it would be useful to create a specific section addressing why the results are not generalizable to similar populations, and this point should be explored in greater depth.
Bibliography: The bibliography should be expanded based on the previous suggestions, and those older than 15-20 years should be updated unless they are of a methodological nature or have strong evidence of impact.
Conclusion: In summary, the manuscript presents scientifically significant findings, but it requires several methodological and structural improvements to enhance its overall quality. My recommendation is to proceed with a thorough revision that addresses these points before moving forward with publication. With appropriate revisions, the manuscript could represent a valuable contribution to the relevant scientific

Reviewer 2 ·

Basic reporting

'no comment'

Experimental design

#1) The type of study and the dependent and independent variables should be clearly explained in the methods section of the abstract and the full text.

#2) The type of statistical test should be stated in the abstract (in the methods section).

Validity of the findings

#1) Biases and how to address them should be clearly stated.

·

Basic reporting

The background is clear and unambiguous. However, I recommend the following:
1. Add some statistics regarding the disease worldwide in general and Taiwan in specific.
2. Pay more attention to grammar.

Experimental design

The research question/goal can be refined to be able to cover the knowledge gap and the importance of this work to fill this gap.

Can you justify the duration of two months?
* A1c is a test that needs to be taken at least every 3 months, can you justify this with the selection of 2 months for the study?
* Normally it takes 60 days (2 months) to acquire a new habit, if step count is newly introduced to the participants, it is best to collect data after two months from adopting the new wearable.
* It would be a good addition to mention the duration of the recruitment and the assessment process.

The methodology is explained clearly and straightforward.

Validity of the findings

The study is not novel in its concept. However, it may be new for the Taiwanese population. The positive impact of physical activity on the management of Type 2 diabetes is well known. Therefore, the study's conclusion is valid.

The reason for the absence of correlation between HbA1c and the increased physical activity could be due to the short duration of the study.

I believe that the patient can serve as his own control in a before-and-after study design.

I have done several studies in diabetes management, whether it was real-time remote monitoring or periodic follow up based on a pre-planned program with the patient.
There is now doubt that any kind of physical activity will indeed help diabetic patients, however, the limitations of the study are tightening the potential of the study.

Reviewer 4 ·

Basic reporting

This manuscript is written in clear manner, easy to grasp the flow of the write up.

References are mostly up to date.

With appropriate revision, I believe this manuscript has potential to be published.

Experimental design

The are few points need to be addressed in regards to the experimental design/protocol. Please refer additional comment section.

Validity of the findings

Revise discussion to avoid overgeneralization.

Additional comments

1. Thank you for the opportunity to read this manuscript. This study addresses an important topic and provides valuable insights into remote patient monitoring (RPM) for older adults with diabetes. However, methodological gaps (e.g., sample size justification, control group) and statistical considerations (e.g., multiplicity adjustment) need addressing to strengthen validity. Below are few comments to further strengthen this manuscript within its limitation.
2. Sample Size Calculation and Study Power. Authors did not mention a sample size calculation or power analysis. A justification for the sample size (N=66) is critical to confirm that the study was adequately powered to detect clinically meaningful effects, especially given the multiple correlations tested. I would suggest to include a post-hoc power analysis or clarify how the sample size was determined (e.g., based on prior studies or feasibility).
3. Sampling Method: Participants were recruited from a single medical center, which may introduce selection bias (e.g., excluding less mobile or rural populations). The exclusion criteria (e.g., cognitive impairment) further limit generalizability to broader populations. I strongly recommend clarifying the recruitment process (e.g., consecutive enrolment, random sampling) and discuss potential biases. Consider acknowledging limitations in generalizability.
4. Statistical Analysis: Non-parametric tests were appropriately used for non-normally distributed data. However, the division of participants into "improved" and "declined" groups lacks clarity (e.g., criteria for grouping, clinical vs. statistical significance).
5. Multiple correlations were tested without adjustment for multiplicity (e.g., Bonferroni correction), increasing the risk of Type I errors. Perhaps authors could define the grouping criteria explicitly and apply corrections for multiple comparisons. Report effect sizes alongside p-values.
6. I found that the absence of a control group makes it difficult to attribute the observed increase in step counts solely to remote monitoring. Confounding factors (e.g., seasonal changes, Hawthorne effect) may influence results. This limitation should be acknowledged and suggest future studies include a control arm (e.g., standard care vs. RPM).
7. Short Follow-Up Period: The 8-week tracking period is insufficient to assess long-term adherence or health outcomes (e.g., glycemic control, frailty progression). Perhaps authors could discuss the short-term nature of the findings and propose extended follow-up in future work.
8. Wearable device accuracy (e.g., Garmin trackers) is not discussed. I would suggest to briefly address device validation or cite prior studies confirming reliability in older populations.
9. Integration of Diabetic P4P Program: The diabetic pay-for-performance (P4P) program’s role in the intervention is unclear. How did it interact with RPM? Could it have confounded results? Further clarification on how the P4P program was integrated and controlled for in analyses.
10. Tables 2 and 3 have formatting issues (e.g., truncated headers, inconsistent decimal places). Figure/table captions are incomplete (e.g., Figure 1 flowchart missing). Please revise tables for clarity and ensure all figures are included and properly labeled. This would help readability of this manuscript.
11. Authors briefly mentions limitations (e.g., selection bias, short follow-up) but does not fully address their implications (e.g., how selection criteria might skew results). Authors should expand the limitations section to discuss how these factors affect interpretation and external validity.
12. Authors speculates on long-term metabolic benefits (e.g., glycemic control) in the discussion section, however, it was done without supporting data. Avoid overclaimd and overgeneralization, instead focus and align with the 8-week findings.

---

## Round 0.2 · Minor Revisions

Dear Dr. Lin,
We suggest you perform a Minor Revision of your manuscript following the instructions of one of the reviewers. The attached document indicates the changes suggested by the reviewer.
Sincerely,
Ana Maria Jiménez-Cebrián

Reviewer 2 ·

Basic reporting

all comments were performed in a good manner.

Experimental design

all comments were performed in a good manner.

Validity of the findings

all comments were performed in a good manner.

·

Basic reporting

Pass

Experimental design

Pass

Validity of the findings

Pass

Additional comments

The authors addressed most of my comments.
However, I would recommend giving points 4 and 5 another go.

Reviewer 4 ·

Basic reporting

Clear

Experimental design

Acceptable.

Although, with a clear lengthy INTERVENTION sub-heading, I might suggest the study design is more toward non-randomised experimental study aka quasi-experimental study. Author stated in the title as pre- and post study anyway.

Validity of the findings

Acceptable

Additional comments

1. Lack of clarity on sample size. I assumed it was determined using G-power analysis. If this indeed what authors used to calculate the study power, it should have clearly mentioned as such. On top of, the chosen effect size required a citation. Upon tried it on myself,

Using t-test family and for the means difference between two dependent group, for post-hoc power analysis on G-power, with the effect size given and alpha level of 0.05, the calculated power was 0.9384 (93.84%). Author reported 92.84.

2. I could not find the statement from author on which type of sampling method they employed. Was it convenient sampling? Purposive sampling? Simple random sampling? Universal sampling? Please address this and indicate in the write up.

3. Addressing reviewer comment in a point by point rebuttal table would be helpful.

Annotated reviews are not available for download in order to protect the identity of reviewers who chose to remain anonymous.

---

## Round 0.3 · accepted · Accept

Dear Dr. Lin,

Thank you for your submission to PeerJ.

I am writing to inform you that your manuscript - Enhancing physical activity in older type 2 diabetic adults through remote patient monitoring: A Pre-Post Study in Taiwan - has been Accepted for publication. Congratulations!

Warm regards,
Ana María Jimenez-Cebrian

Reviewer 4 ·

Basic reporting

Meet the PeerJ requirements

Experimental design

Appropriately stated.

Validity of the findings

Valid.

Additional comments

Thank you for addressing the comments.